# Finite element analysis of a one-piece zirconia implant in anterior single tooth implant applications

Georgi Talmazov[1], Nathan Veilleux[2], Aous Abdulmajeed[1], Sompop Bencharit[1,2,3]*

**1** Department of General Practice, School of Dentistry, Virginia Commonwealth University, Richmond, Virginia, United States of America, **2** Department of Biomedical Engineering, College of Engineering, Virginia Commonwealth University, Richmond, Virginia, United States of America, **3** Department of Oral & Maxillofacial Surgery, School of Dentistry Commonwealth University, Richmond, Virginia, United States of America

* sbencharit@vcu.edu

**Data Availability Statement:** All relevant data are within the manuscript and Supporting Information files.

**Funding:** The authors received no specific funding for this work.

## Abstract

This study evaluated the von Mises stress (MPa) and equivalent strain occurring around monolithic yttria-zirconia (Zir) implant using three clinically simulated finite element analysis (FEA) models for a missing maxillary central incisor. Two unidentified patients' cone-beam computed tomography (CBCT) datasets with and without right maxillary central incisor were used to create the FEA models. Three different FEA models were made with bone structures that represent a healed socket (HS), reduced bone width edentulous site (RB), and immediate extraction socket with graft (EG). A one-piece abutment-implant fixture mimicking Straumann Standard Plus tissue level RN 4.1 X 11.8mm, for titanium alloy (Ti) and Zir were modeled. 178 N oblique load and 200 N vertical load were used to simulate occlusal loading. Von Mises stress and equivalent strain values for around each implant model were measured. Within the HS and RB models the labial-cervical region in the cortical bone exhibited highest stress, with Zir having statistically significant lower stress-strain means than Ti in both labial and palatal aspects. For the EG model the labial-cervical area had no statistically significant difference between Ti and Zir; however, Zir performed better than Ti against the graft. FEA models suggest that Ti, a more elastic material than Zir, contributes to the transduction of more overall forces to the socket compared to Zir. Thus, compared to Ti implants, Zir implants may be less prone to peri-implant bone overloading and subsequent bone loss in high stress areas especially in the labial-cervical region of the cortical bone. Zir implants respond to occlusal loading differently than Ti implants. Zir implants may be more favorable in non-grafted edentulous or immediate extraction with grafting.

## Introduction

Recent demands in esthetics have driven research and clinical applications of monolithic zirconia (Zir) and yttria-stabilized tetragonal zirconia polycrystal ceramics or Y-TZP not only as restorative materials, but to also fabricate dental implants. Zir has proven to have similar

**Competing interests:** The authors have declared that no competing interests exist.

**Abbreviations:** Zir, zirconia; FEA, finite element analysis; CBCT, cone-beam computed tomography; Ti, titanium alloy; STL, standard tessellation language; HS, healed socket with ideal osseous dimension for a single tooth implant; RB, reduced bone width model; EG, extraction site with grafting; Y-TZP, yttria-stabilized Zirconia; Ti6Al4V, Titanium alloy Ti6Al4V; N, Newton.

biological properties and similar short-term survival rate compared to the conventional titanium alloy, Ti-6Al-4V, implants (Ti).[1–3] Beside esthetics, Zir as a restorative material has proven to be superior to other ceramic materials[3–8] in terms of mechanical and biological properties including high fracture toughness, high elastic modulus, low thermal conductivity, as well as low dental plaque affinity. One-piece Zir dental implants have been shown to have clinically acceptable mechanical properties even for a reduced diameter implant design.[7] Placement of Zir dental implants has been advocated for anterior esthetic zone in a conventional healed edentulous site,[1] an immediate extraction site,[9] and immediate provisionalization.[9] Little information currently exists in the literature on the biomechanical relationship between a one-piece Zir dental implant and its peri-implant osseous structure especially in esthetic zone situations in contrast to the availability of clinical data for Ti. There seemed to be a gap in the literature on Zir dental implants in various clinical situations.

Finite element analysis (FEA) has been applied to understand the bone-implant interface, allowing engineers and clinicians to evaluate implant materials, designs, and their effects toward the surrounding osseous structures.[10–14] Implant designs have been developed based on FEA modeling that aids our understanding of the bone-implant interface mechanics especially crestal bone stress distribution where bone loading forces are the highest.[15,16] FEA has long been used to understand the functional loading effects on a single tooth implant [17] in terms of different implant designs, implant-abutment connections, and restorative designs.[18,19] Simulation of various bone types as well as clinical scenarios such as immediate extraction sockets or grafted sites can also be generated and analyzed using FEA models.[20–21] Older FEA studies draw their conclusion from models that have been created using simple geometry (ie. cylinders, rectangular blocks) to simplify computations of the mechanical properties of peri-implant bone and haven't really studied how implants behave in periodontal conditions. Recent studies utilize the advantages of cone beam computed tomography (CBCT) to create high resolution STL meshes used to model the heterogenous peri-implant osseous microstructure and environment improving the accuracy of the model.[21] An FEA study comparing a Ti-Ti two-piece abutment/implant, a Zir abutment/Ti implant, and a one-piece Zir implant, demonstrated that the one-piece Zir implant generated lower stresses in the peri-implant bone region. Marcián et al. concluded that one-piece Zir implants appeared to have favorably less cervical bone stress, implying peri-implant bone preservation in type III bone when compared to Ti control.[22] With esthetic consideration, Zir implant has been indicated in different types of clinical scenarios including healed sockets, immediate implant placement, as well as immediate loading protocol.[1,3,6,9] Note also that anterior edentulous site often has reduced bone width resulting in more challenge in implant placement and in long term maintaining peri-implant bone.[23,24]

This study applied FEA to examine the effects of different clinical scenarios that had not been studied biomechanically. This includes a healed socket (HS), reduced bone width edentulous site (RB), and extraction socket with bone grafting (EG) on the peri-implant osseous structure around a one-piece Zir implant. This study utilizes high resolution segmentation from CBCT to create its simulated osseous structures for the various scenarios studied. The rationale of the study was that the different mechanical properties of Ti and Zir may influence the distribution of von Mises stress and equivalent strain on heterogenous peri-implant osseous structures. The null hypothesis then is that Zir implants loaded with different forces in certain situations would transmit those forces into the surrounding peri-implant bone with similar magnitude and distribution as the Ti implants. By accepting or rejecting the null hypothesis that there is no difference between Ti and Zir on peri-implant bone and graft, the goal of the study is to aid the clinician in establishing a rationale for use of Zir implants in various clinical scenarios a patient may present with in the anterior maxilla.

## Materials and methods

The first step of the workflow (Fig 1) to create the *in silico* three dimensional FEA models was selecting CBCT scans from Virginia Commonwealth University (VCU) School of Dentistry internal radiology database with patient identifiers removed prior to the utilization of the scans. Two CBCT datasets were selected based on the appropriate mesio-distal and facio-lingual dimensions available for an esthetic zone single tooth implant, an edentulous maxillary right central incisor and a maxillary right central incisor that was deemed to be extracted and replaced with a single tooth implant.[25] The CBCT scan protocol used was iCAT FLX V10 (Imaging Sciences International LLC, Hatfield, PA) with standard implant scan parameters (16 deep 10 high cm volume, 0.3 mm voxel size, 8.9-second scan time, 3.7-second exposure time, 120 kVP, 5 mA, and 501.3 mGy/cm2).[26] The CBCT scans were imported into 3D Slicer (https://www.slicer.org/) to generate the initial 3D model[27] surfaces of cortical and trabecular bone through the use of threshold operations and the Grow-Cut algorithm. The maxillary anterior region of interest, area approximate to the maxillary right central incisor was isolated. [25,26,28] The final mesh model was exported as a Standard Tessellation Language (STL) file for final optimization before assembled in the FEA.

The osseous model was subsequently re-meshed and optimized for FEA using MeshLab (http://www.meshlab.net/). While Meshmixer 3.0 software (Autodesk Meshmixer) was used to remove invalid geometry using the Analysis Tool.[29] Three FEA osseous models were generated this way representing the healed socket with ideal osseous dimension for a single tooth implant (HS), a reduced bone width model (RB), and an extraction site with grafting (EG). These FEA models were built mimicking the clinical scenarios commonly found for a single tooth replacement for a maxillary first central incisor.[30,31] The HS and RB models were created from the partially edentulous CBCT dataset, while the EG model was created from the dentate CBCT dataset.

The mesh was optimized using MeshLab in order to improve accuracy and speed of the computational analysis.[32] Within MeshLab, each model mesh was simplified by using quadric edge collapse decimation function to reduce the total number of initial mesh nodes to 543,804 for the HS, 704,960 for the RB, and 766,888 for the EG models. Cortical bone mesh was modified to match the appropriate bone width for the chosen implant using Meshmixer. Building upon the HS model, the Siebert Class 1 defect mesh in the RB model was created by modifying the cortical bone layer in Autodesk Meshmixer. The residual ridge defect commonly seen clinically on the labial aspect of the implant was created (Fig 2). The EG model or

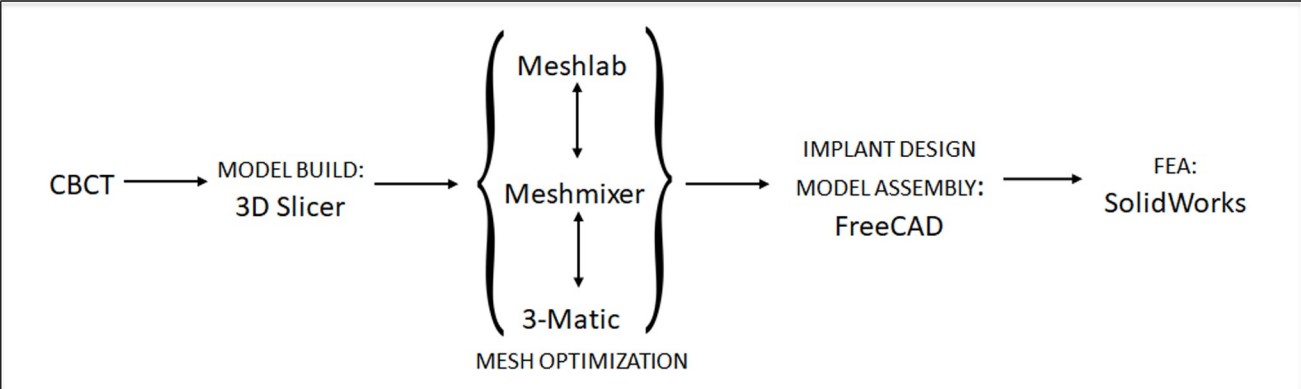

**Fig 1. FEA modeling workflow.**

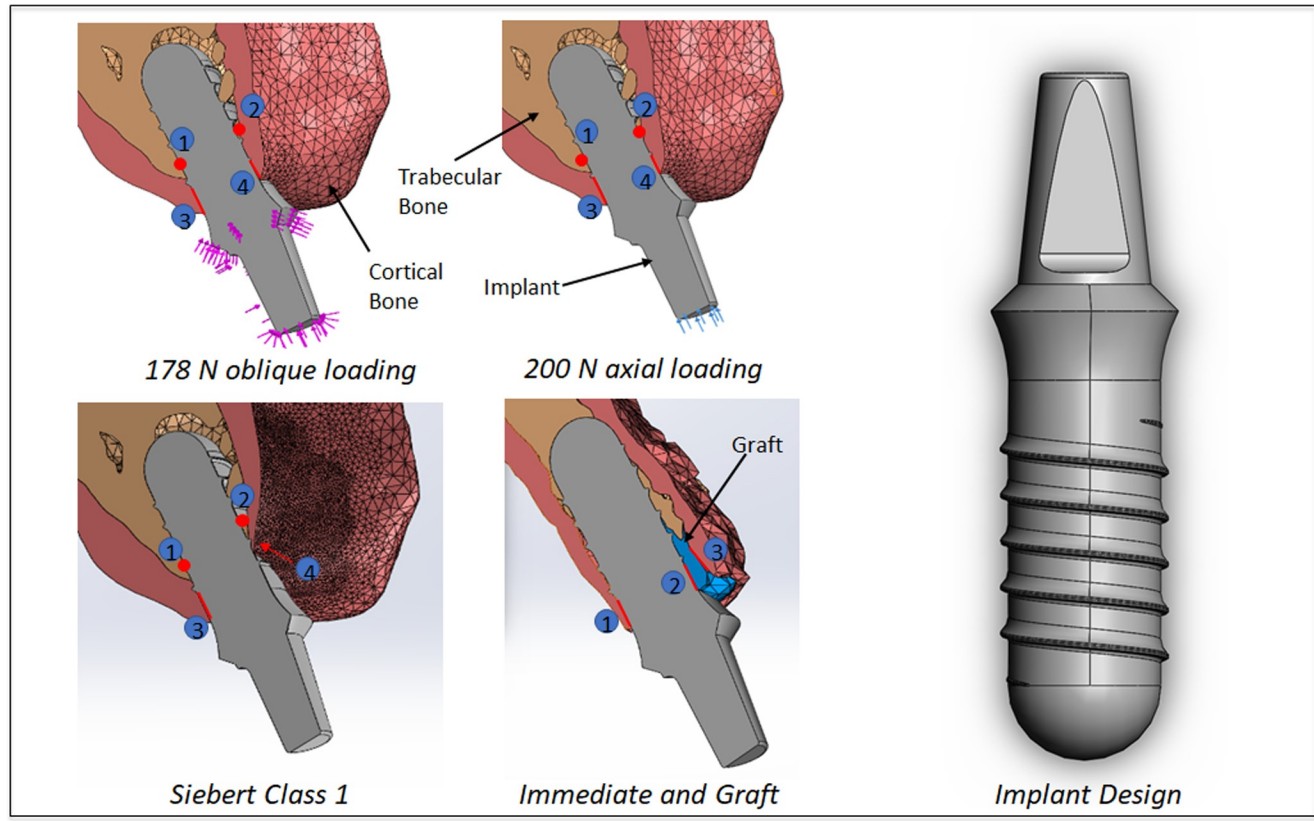

**Fig 2. FEA model, loading protocols, measurement sites, and implant design.**

immediate implant placement and grafting socket model was developed from the dentate CBCT dataset.

A tissue-level root-form endosseous dental implant (Straumann Standard Plus Tissue Level RN) was used as an implant model similar to previous studies.[33,34] The model was generated using FreeCAD. (https://www.freecadweb.org/) [35] The implant had a 0.2 mm thread depth and 1.2mm pitch, carrying a 3.9 mm inner diameter, and osseous height of 10 mm and 1.8 mm platform height, from bone crest, for the tissues. It was modeled with cervical diameter of 4.8 mm (regular neck, RN, Straumann), fixture diameter of 4.1 mm, and a 5.5 mm in height anti-rotation abutment component. The analysis was focused on the one-piece Zir implant and therefore the abutment and implant were unified as one solid unit.[36] For the HS and RB models, the implant fixture was optimally positioned by taking into consideration buccal-palatal width, angulation, and expected soft tissue thickness & emergence profile (Fig 2). For the EG model, the implant fixture was placed within the tooth socket to match the angulation and emergence profile of other models while keeping the implant engagement to the palatal bone socket wall.[25,28,37] The remaining void in the socket between the implant and the labial bone was filled with mesh pieces to simulate the grafted material (Fig 2).

Each bone model underwent a final mesh refinement in 3-Matic (Materialise NV) for further triangle reduction and removal of unconnected, floating bone segments. The wrap operation in 3-Matic was used to remove sharp edges within each surface, a requirement to allow the mesh to be used in a FEA. Each model's bone parts and implant were then imported and combined in Solidworks (DS Solidworks Corp). The physical properties were applied and the simulations were carried out in Solidworks.[38,39] Cortical, trabecular, and morselized

**Table 1. List of materials used in this finite element analysis, their properties and numeric values used for the simulation[*].**

|  | Y-TZP[52] | Ti-6Al-4V | Cortical Bone | Trabecular Bone | Morselized Cancellous Bone[53] |
|---|---|---|---|---|---|
| Elastic Modulus (MPa) | 220000[52] | 104800.31[7] | 17000[4] | 2000[9] | 100[53] |
| Poisson's Ratio | 0.31[52] | 0.31[7] | 0.42[6] | 0.4[10] | 0.2[53] |
| Mass Density (kg/m^3) | 6000[52] | 4428.78[7] | 1900[8] | 270[2] | 340[1] |
| Tensile Strength (MPa) | 745[52] | 1050[7] | 52[4] |  |  |
| Compressive Strength (MPa) | 2200[52] | 848–1080[5] | 130[4] | 104[3] |  |
| Yield Strength (MPa) | 300[52] | 827.37088[7] | 49[6] | 4.8[3] | 50[1] |

[*]The empty cells indicate missing values that were not found in the literature and were not required for the FEA to compute. When available, the values were used to improve the accuracy of the model.

cancellous bone material properties were applied to the appropriate components (Table 1). The implant materials used were Yttria-stabilized Zirconia (Y-TZP) and Titanium alloy (Ti-6Al-4V), created in two separate study simulations for each model. Fixations were applied at the palatal direction of the abutment portion of the model. Then for each study comparing the two materials, two loading conditions were applied: 200 N force directed down the long axis and 178 N force directed along the implant labial axial wall, implant-crown margins, and incisal surface (Fig 3A and 3B). The force values were simulated normal biting force.[40,41] The

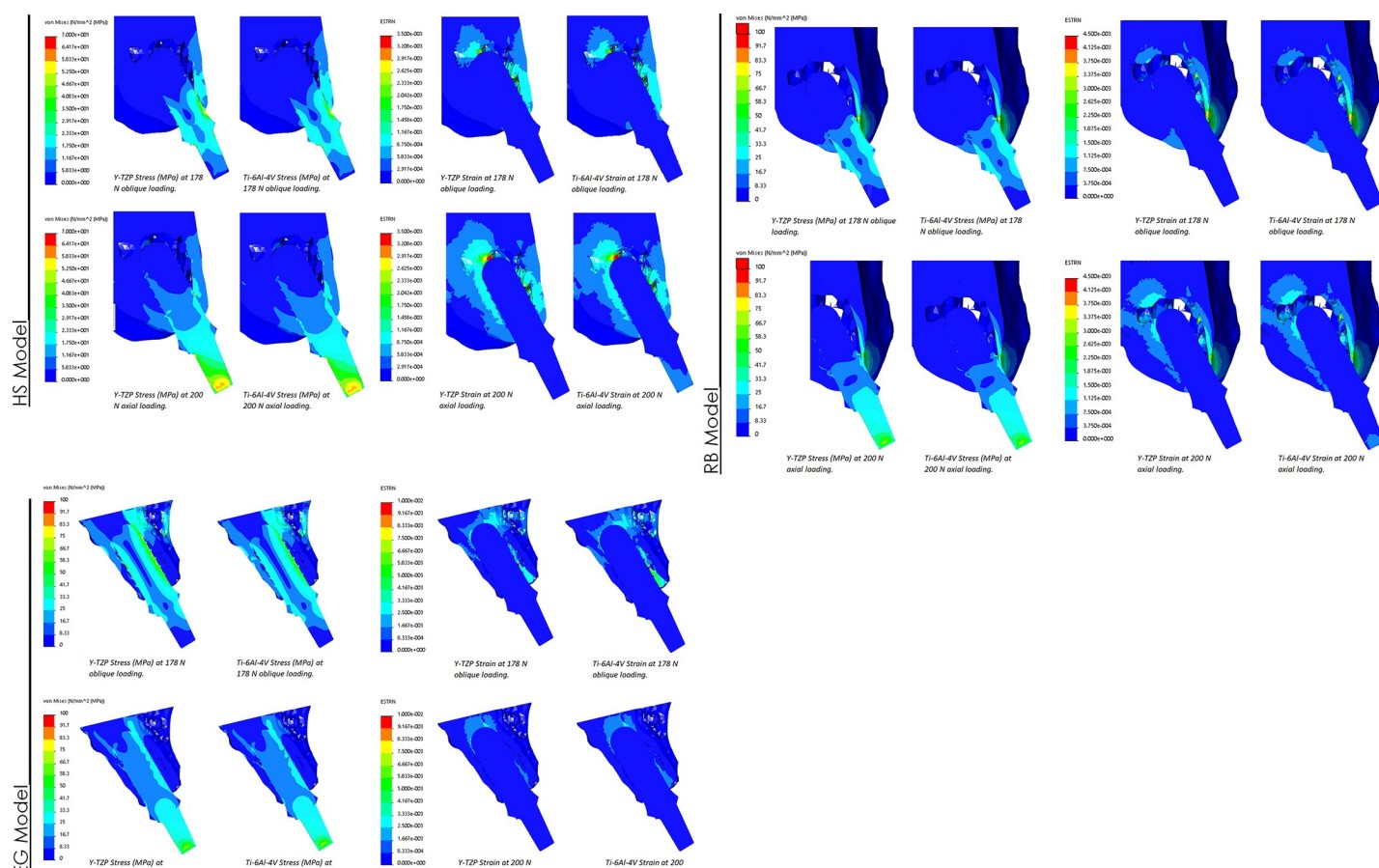

**Fig 3. FEA model and loading. (A)** HS Model Stress and Strain Distributions, **(B)** RB Model Stress and Strain Distributions, and **(C)** EG Model Stress and Strain Distributions.

oblique loading is a component of axial loading and perpendicular to long axis force application resulting in torque. The three regions (Fig 3A and 3B) were identified to simulate the sum of contact points for a single unit crown in an Angle's Class I occlusion. All the models have an assumption of bonded contact at all bone interfaces as well as at the implant-bone interfaces presuming complete osseointegration. The von Mises stress and equivalent strain distribution along the implant-bone interface were quantitatively reviewed. Von Mises stress and equivalent strain means were obtained from measurements of 10 (n = 10) fixated sensors across all models within the designated regions (Figs 2 and 3). From the sampled data in cortical and trabecular bone at implant interface across the 3 models, 2 loading scenarios, and 2 implant materials means and respective standard deviations were calculated in Excel (Microsoft Excel). To compare the biomechanical significance between Ti and Zir implants one-tailed p-values with $\alpha = 0.05$ were computed from the 10 sample points obtained from regions 1–4 (Fig 2) for von Mises stress and equivalent strain.

## Results

Data measurements along the implants surfaces at the designated regions (Fig 2) from the FEA analysis showed that overall a Zir implant has higher internal von Mises stress (MPa) and lower equivalent strain when compared to its Ti counterpart (Table 2 and Table 3; Fig 4). Concurrently, for the HS model there was p-value significance of less than 0.01 for the 178 N oblique loading. For 200 N axial loading, Zir had significantly higher stress labially with p-value of

**Table 2. Stresses and strains computed from three FEA models at 178 N oblique loading.**

| HS MODEL | | | | | | | | | |
|---|---|---|---|---|---|---|---|---|---|
| **Stress (MPa)** | | | | | **Strain** | | | | |
| | | **Ti-6Al-4V** | **Y-TZP** | **P-value** | | | **Ti-6Al-4V** | **Y-TZP** | **P-value** |
| Labial | Implant (S4) | 39.18±9.84 | 43.18±8.13 | 0.0004 | Labial | Implant (S4) | 3.51E-04±2.59E-05 | 1.80E-04±6.55E-06 | 4E-10 |
| | Cortical (S4) | 13.3±5.91 | 10.4±3.82 | 0.002 | | Cortical (S4) | 7.87E-04±1.53E-04 | 6.09E-04±8.40E-05 | 2E-05 |
| | Trabecular (S2) | 3.90±1.02 | 4.05±1.16 | 0.01 | | Trabecular (S2) | 1.39E-03±5.01E-04 | 1.43E-03±5.40E-04 | 0.01 |
| Palatal | Implant (S3) | 22.1±7.36 | 25.58±7.33 | 2E-05 | Palatal | Implant (S3) | 1.94E-04±1.73E-05 | 1.06E-04±4.63E-06 | 3E-09 |
| | Cortical (S3) | 6.15±4.51 | 4.28±2.47 | 0.009 | | Cortical (S3) | 3.31E-04±1.05E-04 | 2.36E-04±5.49E-05 | 0.0001 |
| | Trabecular (S1) | 0.22±0.0638 | 0.19±0.0809 | 0.2 | | Trabecular (S1) | 1.18E-04±3.38E-05 | 1.12E-04±9.81E-06 | 0.3 |
| RB MODEL | | | | | | | | | |
| Stress (MPa) | | | | | Strain | | | | |
| | | Ti-6Al-4V | Y-TZP | P-value | | | Ti-6Al-4V | Y-TZP | P-value |
| Labial | Implant (S4) | 45.93±9.21 | 47.2±10.3 | 0.08 | Labial | Implant (S4) | 3.89E-04±2.46E-05 | 1.89E-04±1.22E-05 | 1E-12 |
| | Cortical (S4) | 48.6±18.8 | 45.9±20.7 | 0.04 | | Cortical (S4) | 1.71E-03±6.51E-04 | 1.46E-03±9.04E-04 | 0.008 |
| | Trabecular (S2) | 0.311±0.225 | 0.227±0.13 | 0.01 | | Trabecular (S2) | 1.77E-04±9.77E-05 | 1.30E-04±5.61E-05 | 0.003 |
| Palatal | Implant (S3) | 20.51±10.6 | 22.689±10.4 | 0.0003 | Palatal | Implant (S3) | 1.92E-04±3.86E-05 | 9.97E-05±1.63E-05 | 2E-07 |
| | Cortical (S3) | 10.24±7.21 | 9.11±5.22 | 0.07 | | Cortical (S3) | 6.86E-04±2.57E-04 | 5.98E-04±1.75E-04 | 0.004 |
| | Trabecular (S1) | 0.38±0.129 | 0.36±0.134 | 0.01 | | Trabecular (S1) | 1.50E-04±4.92E-05 | 1.65E-04±4.72E-05 | 0.0002 |
| EG MODEL | | | | | | | | | |
| Stress (MPa) | | | | | Strain | | | | |
| | | Ti-6Al-4V | Y-TZP | P-value | | | Ti-6Al-4V | Y-TZP | P-value |
| Labial | Implant (S2) | 55.41±4.56 | 57.34±5.20 | 8E-06 | Labial | Implant (S2) | 4.75E-04±2.51E-05 | 2.36E-04±1.27E-05 | 2E-13 |
| | Cortical (S3) | 3.44±0.544 | 3.17±0.997 | 0.1 | | Cortical (S3) | 1.44E-04±2.06E-05 | 1.31E-04±4.36E-05 | 0.09 |
| | Graft (S2) | 0.566±0.124 | 0.429±0.0844 | 1E-06 | | Graft (S2) | 4.44E-03±7.55E-04 | 3.36E-03±5.10E-04 | 1E-07 |
| Palatal | Implant (S1) | 40.46±1.95 | 42.18±4.13 | 0.04 | Palatal | Implant (S1) | 3.24E-04±8.39E-06 | 1.68E-04±8.17E-06 | 7E-19 |
| | Cortical (S1) | 11.5±5.65 | 6.83±2.87 | 0.0002 | | Cortical (S1) | 5.24E-04±3.11E-05 | 3.18E-04±2.80E-05 | 8E-17 |

**Table 3. Stresses and strains computed from three FEA models at 200 N loading along implant axis.**

| HS MODEL | | | | | | | | | |
|---|---|---|---|---|---|---|---|---|---|
| Stress (MPa) | | | | | Strain | | | | |
| | | Ti-6Al-4V | Y-TZP | P-value | | | Ti-6Al-4V | Y-TZP | P-value |
| Labial | Implant (S4) | 17.88±2.52 | 18.81±2.96 | 0.03 | Labial | Implant (S4) | 1.41E-04±9.37E-06 | 6.89E-05±5.71E-06 | 4E-12 |
| | Cortical (S4) | 9.48±2.53 | 8.26±2.15 | 0.0003 | | Cortical (S4) | 5.12E-04±5.31E-05 | 4.50E-04±4.32E-05 | 5E-06 |
| | Trabecular (S2) | 4.32±0.966 | 4.12±1.05 | 0.002 | | Trabecular (S2) | 1.54E-03±5.52E-04 | 1.47E-03±5.52E-04 | 4E-05 |
| Palatal | Implant (S3) | 12.0±3.5 | 12.7±3.5 | 0.006 | Palatal | Implant (S3) | 1.03E-04±6.54E-06 | 5.11E-05±2.71E-06 | 2E-11 |
| | Cortical (S3) | 8.44±3.77 | 7.76±2.94 | 0.04 | | Cortical (S3) | 4.48E-04±6.88E-05 | 4.15E-04±5.20E-05 | 0.0001 |
| | Trabecular (S1) | 0.85±0.502 | 0.78±0.572 | 0.02 | | Trabecular (S1) | 5.12E-04±6.46E-05 | 4.88E-04±9.00E-05 | 0.01 |
| **RB MODEL** | | | | | | | | | |
| Stress (MPa) | | | | | Strain | | | | |
| | | Ti-6Al-4V | Y-TZP | P-value | | | Ti-6Al-4V | Y-TZP | P-value |
| Labial | Implant (S4) | 23.15±9.90 | 23.11±10.62 | 0.5 | Labial | Implant (S4) | 2.24E-04±4.25E-05 | 1.05E-04±2.11E-05 | 1E-08 |
| | Cortical (S4) | 39.9±16.5 | 37.7±17.6 | 0.04 | | Cortical (S4) | 1.30E-03±5.28E-04 | 1.13E-03±7.10E-04 | 0.01 |
| | Trabecular (S2) | 0.394±0.208 | 0.273±0.11 | 0.004 | | Trabecular (S2) | 2.17E-04±9.33E-05 | 1.42E-04±3.78E-05 | 0.001 |
| Palatal | Implant (S3) | 12.02±3.23 | 12.82±3.68 | 0.01 | Palatal | Implant (S3) | 1.02E-04±6.02E-06 | 5.15E-05±4.00E-06 | 1E-13 |
| | Cortical (S3) | 8.22±2.79 | 7.50±2.27 | 0.004 | | Cortical (S3) | 4.57E-04±9.74E-05 | 4.18E-04±7.65E-05 | 0.0001 |
| | Trabecular (S1) | 0.78±0.271 | 0.73±0.283 | 0.002 | | Trabecular (S1) | 4.13E-04±8.60E-05 | 3.86E-04±8.65E-05 | 0.00004 |
| **EG MODEL** | | | | | | | | | |
| Stress (MPa) | | | | | Strain | | | | |
| | | Ti-6Al-4V | Y-TZP | P-value | | | Ti-6Al-4V | Y-TZP | P-value |
| Labial | Implant (S2) | 18.31±1.09 | 18.13±1.02 | 8E-05 | Labial | Implant (S2) | 1.54E-04±6.95E-06 | 7.27E-05±3.19E-06 | 7E-14 |
| | Cortical (S3) | 0.639±0.196 | 0.609±0.264 | 0.2 | | Cortical (S3) | 2.69E-05±8.52E-06 | 2.56E-05±1.17E-05 | 0.2 |
| | Graft (S2) | 0.148±0.0318 | 0.105±0.0209 | 3E-07 | | Graft (S2) | 1.16E-03±1.96E-04 | 8.26E-04±1.29E-04 | 4E-08 |
| Palatal | Implant (S1) | 15.80±1.84 | 16.24±1.16 | 0.07 | Palatal | Implant (S1) | 1.36E-04±1.14E-05 | 6.51E-05±3.18E-06 | 3E-10 |
| | Cortical (S1) | 3.82±3.22 | 2.13±1.92 | 0.001 | | Cortical (S1) | 1.50E-04±7.34E-06 | 8.17E-05±5.53E-06 | 9E-15 |

0.03 and less than 0.01 palatally. Strain was significantly lower for Zir on labial and palatal aspects with p-values less than 0.01. The RB model's Ti implant surface had significantly higher equivalent strain for 178 N oblique and 200 N axial loading, with labial and palatal p-value comparisons less than 0.01. For this model the labial internal von Mises stress difference was not significant for either loading conditions, p-values greater than 0.05. While the palatal for both loading conditions was greater for Zir, p-value less than 0.01 for 178 N oblique and 0.01 for 200 N axial. For the EG model, 178 N oblique loading showed higher von Mises stress for Zir where labial p-value was less than 0.01 and palatal was 0.04. Long axial loading had p-value less than 0.01 labially but Ti had higher stress in that region. Palatal stress difference was not significant (p-value 0.07). With respect to internal strain, Ti exhibited higher strain for both loading condition and it was significantly different compared to Ti with p-values less than 0.01 for all fixture regions.

The FEA models' simulated bone samples of von Mises stress and equivalent strain were produced by loading and implant fixture of Ti and Zir vertically at 200 N and obliquely at 178 N. The distributions of von Mises stress and equivalent strain mean, standard deviation, and p-values were computed (Table 2 and Table 3; and Fig 4). From the HS model under 178 N oblique loading the labial aspect of the cortical bone showed a mean von Mises stress of 13.3 ±5.91 MPa for Ti and 10.4±3.82 MPa for Zir while on the palatal aspect 6.15±4.51 MPa and 4.28±2.47 MPa were observed respectively for Ti and Zir, both regions with a p-value of less than 0.01. For HS model with 178 N oblique loading labial area equivalent strain mean was

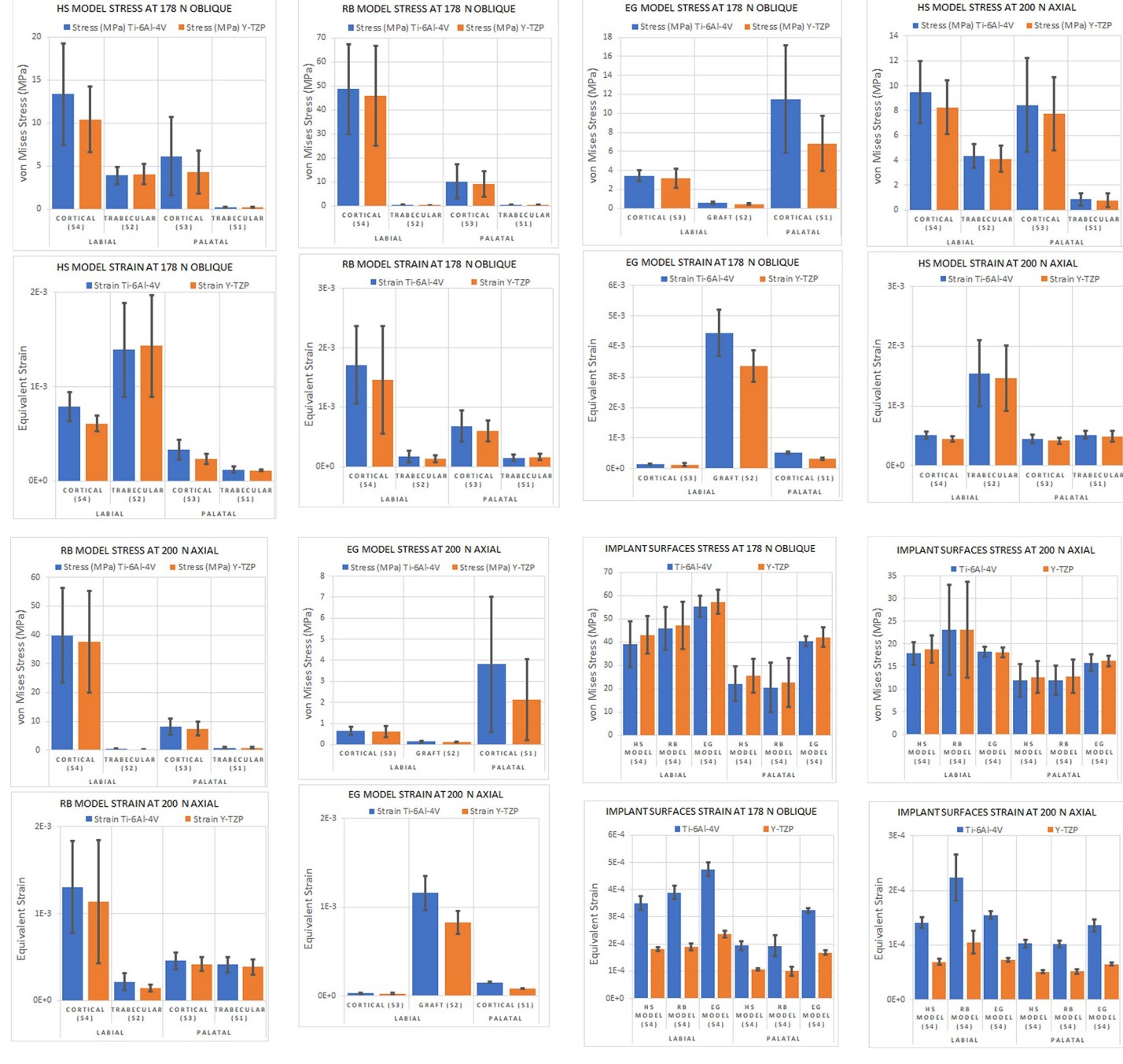

**Fig 4.** Model loading at 178 N oblique for HS (4A), RB (4B), and EG (4C); at 200 N vertical for HS (4D), RB (4E), and EG (4F); and the summary of the two loading protocols (4G) and (4H).

found to be $7.87 \times 10^{-4} \pm 1.53 \times 10^{-4}$ for Ti while $6.09 \times 10^{-4} \pm 8.40 \times 10^{-5}$ for Zir. Whilst palatal region for Ti exhibited mean equivalent strain of $3.31 \times 10^{-4} \pm 1.05 \times 10^{-4}$ and $2.36 \times 10^{-4} \pm 5.49 \times 10^{-5}$ for Zir. Labial and palatal equivalent strain p-values were less than 0.01. Trabecular bone exhibited overall lower mean von Mises stress and equivalent strain compared to its cortical counterpart (Tables 2 and 3). Trabecular bone von Mises stress p-value for the labial aspect was 0.01 and palatal was 0.25 –implying significant difference between Ti and Zir, favoring Ti, in

compression but no significance in tension. Trabecular bone equivalent strain labial region p-value was 0.01 implying statistical significance favoring Ti, while the palatal region p-value was 0.28 showing no statistical significance. With respect to long axial loading at 200 N, the mean von Mises stress at the labial cortical interface was 9.48±2.53 MPa for Ti and 8.26±2.15 MPa for Zir while on the palatal cortical interface was 8.44±3.77 MPa for Ti and 7.76±2.94 MPa for Zir. Labial p-value between the two materials was found to be less than 0.01and the palatal p-value was 0.04 showing statistical significance in von Mises stress between Ti and Zir. Analysis of the cortical labial equivalent strain showed a mean of $5.12 \times 10^{-4} \pm 5.31 \times 10^{-5}$ for Ti and $4.50 \times 10^{-4} \pm 4.32 \times 10^{-5}$ for Zir while the palatal aspect had a mean of $4.48 \times 10^{-4} \pm 6.88 \times 10^{-5}$ and $4.15 \times 10^{-4} \pm 5.20 \times 10^{-5}$, respectively. The equivalent strain p-values were less than 0.01 for labial and palatal aspects–implying statistical significance in both regions between Ti and Zir. With respect to the trabecular bone von Mises stress labial p-value was calculated to be less than 0.01 and palatal 0.02, while the equivalent strain p-values were both less than 0.01. There is statistical significance between Ti and Zir within trabecular bone at the labial and palatal aspects of the HS model that favor Zir.

The RB model showed cortical mean von Mises stress on the labial aspect under 178 N oblique loading of 48.6±18.8 MPa for Ti and 45.9±20.7 MPa for Zir, while on the palatal aspect those values were 10.24±7.21 MPa and 9.11±5.22 MPa respectively. The equivalent strain was found to be $1.71 \times 10^{-3} \pm 6.51 \times 10^{-4}$ and $1.46 \times 10^{-3} \pm 9.04 \times 10^{-4}$ respectively for the palatal region, and $6.86 \times 10^{-4} \pm 2.57 \times 10^{-4}$ and $5.98 \times 10^{-4} \pm 1.75 \times 10^{-4}$ respectively. The calculated von Mises stress p-value for the labial region was 0.04 and the palatal 0.07, while the equivalent strain comparison has a p-value less than 0.01 for the labial and palatal. For the trabecular bone, the von Mises stress p-value between Ti and Zir was calculated to be 0.01 for the labial and palatal regions–Ti had significantly higher stress than Zir. P-value calculated for trabecular bone equivalent strain was less than 0.01 for labial and palatal–Ti had significantly higher strain than Zir on the labial aspect but the opposite occurred for the palatal. With respect to the 200 N long axial loading condition, the von Mises stress in labial cortical bone was 39.9±16.5 MPa for Ti and 37.7±17.6 MPa for Zir, while the palatal region showed 8.22±2.79 MPa and 7.50 ±2.27 MPa, respectively. Associated p-values for stress within the labial region was found to be 0.04 and less than 0.01 for the palatal–Ti showed significantly higher stress in both regions. Equivalent strain wise a similar pattern appeared–labial cortical mean was $1.30 \times 10^{-3} \pm 5.28 \times 10^{-4}$ for Ti and $1.13 \times 10^{-3} \pm 7.10 \times 10^{-4}$ for Zir, while palatal was $4.57 \times 10^{-4} \pm 9.74 \times 10^{-5}$ for Ti and $4.18 \times 10^{-4} \pm 7.65 \times 10^{-5}$ for Zir. P-values were found to be 0.01 and less than 0.01 respectively. In trabecular bone, the von Mises stress as well as equivalent strain were significantly higher for Ti in both labial and palatal aspects–p-values both less than 0.01 for stress and strain.

The EG model overall favored Zir where Ti exhibited higher mean values in von Mises stress as well as equivalent strain. Under oblique loading at 178 N the labial cortical bone adjacent to the graft interface had a mean von Mises stress of 3.44±0.544 MPa for Ti and 3.17 ±0.997 MPa for Zir; however, with a p-value of 0.09 this was not statistically significant. The palatal region interfacing the implant surface had a mean von Mises stress of 11.50±5.65 MPa for Ti and 6.83±2.87 MPa for Zir–p-value of less than 0.01 was calculated signifying statistical significance. The stress analysis at the implant-graft interface showed a mean of 0.566±0.124 MPa for Ti and 0.429±0.0844 MPa, calculated p-value was less than 0.01. Equivalent strain wise the labial aspect had a mean of $1.44 \times 10^{-4} \pm 0.206 \times 10^{-4}$ for Ti and $1.31 \times 10^{-4} \pm 0.436 \times 10^{-4}$, and palatally $5.24 \times 10^{-4} \pm 0.311 \times 10^{-4}$ for Ti and $3.18 \times 10^{-4} \pm 0.280 \times 10^{-4}$ for Zir. Calculated p-values from the sample points were 0.09 and less than 0.01, respectively. At the graft-implant interface the equivalent strain for Ti was $4.44 \times 10^{-3} \pm 0.755 \times 10^{-3}$ and $3.36 \times 10^{-3} \pm 0.51 \times 10^{-3}$ for Zir with p-value of less than 0.01. For 200 N long axial loading the same pattern of stress and strain was observed such that Zir was favored over Ti.

## Discussion

Zir dental implants have recently been introduced and advocated for the esthetic zone.[1,3,6] This study was one of the first to apply FEA models to represent the use of a Zir single implant in three common clinical scenarios in the esthetic zone; the healed socket or edentulous area (HS model), in the reduced bone width area (RB model), and in the extraction socket with grafting (EG model). Since the modulus of elasticity of Zir is approximately twice that of Ti, it was hypothesized that the stress/strain distribution might have been different. The results suggest that the stress/strain distributions were very similar between the two materials but were significantly different. Zir demonstrated lower von Mises stress and equivalent strain mean values in all measurements across all models' cortical bone. These results are similar to a previous FEA study comparing one-piece Zir implants with two-piece Zir/Ti abutment/implants, and two-piece Ti/Ti abutment/implants.[22] The higher elastic modulus of Zir provides for a "stiffer" material that dissipates more stress within its internal structure and equivalently lower displacement (ie. strain) of surrounding biologic structures–reflected in the implant's high internal von Mises stress and low equivalent strain (Fig 4). This shows that use of Zir in the three scenarios studied can potentially improve clinical outcomes and reduce peri-implant bone loss especially in the cervical area where bone resorption occurs more readily.[21,22] A recent FEA implant model study pointed out a phenomenon that zirconia received higher stress value compared to titanium.[42] The authors further suggested that there may be a direct relationship between Young's modulus of the material and the stress transferred to the implant, that a more rigid implant absorbs more stress.[42] This may partly explain why in our study, there was less stress in the peri-implant bone in the Zir implant compared to the Ti implant.

In this study, the CBCT scans were used to provide microstructures of the trabecular bone mimicking the natural bone similar to others.[21,43–46] These CBCT based FEA models allow von Mises stress and equivalent stain distributions to be simulated more realistically than other geometrically simplified FEA models.[10,21,46] More importantly, the three FEA models, the healed edentulous site (HS model), reduced bone width edentulous site (RB model), and immediate implant placement with grafting (EB model), allow for a direct comparison between Zir and Ti implants as well as the examination of Zir in common scenarios. Zir implants provide an overall more favorable peri-implant bone stress/strain distribution. This suggests that in non-grafted edentulous single tooth implant site, Seibert defects that experience compression, and the grafted extraction socket site, Zir implants are mechanically superior to Ti implants in terms of occlusal load distribution in the oblique and axial direction. The EG model results suggest Zir implants may allow less graft resorption/labial bone resorption for immediate placement as well as their use in sites with periodontal defects. However, Zir implants and Ti implants in the immediate placement cases may also be selected based on the other clinical scenarios/esthetic considerations. A recent systematic review suggest that Zir implants have similar short time survival rate as Ti implants, ~92%.[1] Another systematic review showed Zir may be an alternative implant material for Ti.[3] On the contrary, one systematic review pointed out the low evidence level of Zir implant studies and advised caution in clinical use of Zir implants.[6] In the immediate implant scenario higher insertion torque, representing good primary stability, may be easier to achieve in the Ti implant due to their lower elastic modulus. Therefore, fracture of Zir implants can be an issue especially in a two-piece or small diameter Zir implants when higher torque is applied.[7]

This study shares similar limitations with other FEA studies. These limitations include the limited study designs, material properties, occlusal loads, individual variations.[10] This study only targets three scenarios designed to compare two materials with one-piece implant

designs. The results may not be applicable to two-piece implants.[22,46] The material properties used here are the ones commonly prescribed in FEA studies.[22] The particular properties may be an average and can vary in individual patients. The occlusal loads applied here in the FEA models is to provide the general idea of how stress/strain distributions would be.[47] The models unfortunately may not represent an individual's physiologic bite and consequently occlusal loading.[48] Assessing various patient-specific bone properties and occlusal loading is unrealistic. Finally, while the CBCT scans from a human volunteer were chosen here, the scans may not be a representative of a wide-range of clinical cases.[43–46] It is possible to define the properties of bone directly based off of the CT intensity and use more subjects; however, the technique requires more specific computational modeling and the additional value of the information for stress evaluation is not clear.[49]

Zir implants demonstrate a promising potential when compared to Ti implants, and based on the conducted FEA analysis the null hypothesis has been rejected. However, clinicians should consider other clinical parameters, such as esthetic demands, insertion torque, clinical occlusion, parafunctional habits, position of the site, type of edentulous areas, etc.[50–54] A prospective clinical study with a large population will be needed to prove that Zir implants are a good clinical alternative to Ti implants. Furthermore, a supplemental *in vitro* analysis to support the computational models is subsequently employed to verify the *in silico* results–FEA models are meant to show how materials behave and not be used for quantitative experimental data.

## Conclusions

In general Zir implants behave more favorable then Ti implants in terms of peri-implant stress distributions. Three different FEA models, healed edentulous site (HS), vertical periodontal defect under compression (RB), and immediate extraction with bone grafting site (EG), mimicking the common clinical scenarios suggested the following conclusion. Due to the stiffness of the material and its intrinsically high elastic modulus, Zir implants transmit less von Mises stress and induce lower equivalent strain to the peri-implant bone compared to Ti implants. This was statistically significant when cervical cortical bone (HS and RB models) and graft (EG model) were evaluated for one-piece Zir implant compared to its Ti counterpart. Therefore, the peri-implant bone surrounding Zir implants may be less prone to mechanically induced biologic peri-implant bone resorption. Zir implants may be considered not only due to its esthetic properties, but also due to the stress modulation properties of the material.

## Supporting information

**S1 Data.**
(XLSX)

## Acknowledgments

The authors thank Dr. Jennifer Wayne and her laboratory for the assistance in the FEA modeling software. Thanks also to the faculty and staff of the VCU Center of Digital Dentistry.

## Author Contributions

**Conceptualization:** Georgi Talmazov, Nathan Veilleux, Aous Abdulmajeed, Sompop Bencharit.

**Data curation:** Georgi Talmazov.

**Formal analysis:** Georgi Talmazov, Nathan Veilleux.

**Investigation:** Georgi Talmazov, Nathan Veilleux.

**Methodology:** Georgi Talmazov, Aous Abdulmajeed, Sompop Bencharit.

**Project administration:** Sompop Bencharit.

**Software:** Georgi Talmazov.

**Supervision:** Sompop Bencharit.

**Writing – original draft:** Georgi Talmazov, Nathan Veilleux, Aous Abdulmajeed, Sompop Bencharit.

**Writing – review & editing:** Georgi Talmazov, Sompop Bencharit.

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
