## [Decision Letter · Decision Letter 0]

22 Nov 2019

PONE-D-19-30284

Finite Element Analysis of a One-piece Zirconia Implant in Anterior Single Tooth Implant Applications

PLOS ONE

Dear Dr. Bencharit,

Thank you for submitting your manuscript to PLOS ONE. After careful consideration, we feel that it has merit but does not fully meet PLOS ONE’s publication criteria as it currently stands. Therefore, we invite you to submit a revised version of the manuscript that addresses the points raised during the review process.

Please, address all the comments made by the reviewers, specially those pointed out by Reviewers #1 and #3.

We would appreciate receiving your revised manuscript by Jan 06 2020 11:59PM. To enhance the reproducibility of your results, we recommend that if applicable you deposit your laboratory protocols in protocols.io, where a protocol can be assigned its own identifier (DOI) such that it can be cited independently in the future. For instructions see: http://journals.plos.org/plosone/s/submission-guidelines#loc-laboratory-protocols

We look forward to receiving your revised manuscript.

Kind regards,

Antonio Riveiro Rodríguez, PhD

Academic Editor

PLOS ONE

2. In your methods section you write that the study used CBCT scans from an implant clinic database with patient identifiers removed. Please provide the name of the database and clarify whether the data were anonymized or de-identified before you accessed them.

- In your data availability statement you write, "All relevant data are within the paper." Please ensure you have provided the individual data points used to create the figures and determine means, medians and variance measures presented in the results, tables and figures and provided details of the database from which the underlying images were obtained such that other researchers can replicate the analyses (http://journals.plos.org/plosone/s/data-availability#loc-faqs-for-data-policy).

4.  We note that Figure(s) [#] in your submission contain copyrighted images. All PLOS content is published under the Creative Commons Attribution License (CC BY 4.0), which means that the manuscript, images, and Supporting Information files will be freely available online, and any third party is permitted to access, download, copy, distribute, and use these materials in any way, even commercially, with proper attribution. For more information, see our copyright guidelines: http://journals.plos.org/plosone/s/licenses-and-copyright.

1.         You may seek permission from the original copyright holder of Figure(s) [#] to publish the content specifically under the CC BY 4.0 license.

Reviewers' comments:

Reviewer's Responses to Questions

**Comments to the Author**

1. Is the manuscript technically sound, and do the data support the conclusions?

Reviewer #1: No

Reviewer #2: Yes

Reviewer #3: Partly

2. Has the statistical analysis been performed appropriately and rigorously? 

Reviewer #1: No

Reviewer #2: N/A

Reviewer #3: No

3. Have the authors made all data underlying the findings in their manuscript fully available?

Reviewer #1: No

Reviewer #2: Yes

Reviewer #3: Yes

4. Is the manuscript presented in an intelligible fashion and written in standard English?

Reviewer #1: No

Reviewer #2: Yes

Reviewer #3: Yes

5. Review Comments to the Author

Reviewer #1: This article numerically investigates one-piece zirconia implant in anterior single tooth implant applications. The paper presented in a poor quality. The results are NOT clearly explained. The structure of the paper does NOT follow any specific method. From reviewer's point of view, the article is required more than a major revision. In the following, there are comments that are appropriate to reinforce the article.

- Some of the most important research results should be mentioned at the end of this section.

- The paper needs keywords.

- The subtitles should be numbered.

- The quality of the figures should be increased.

- The graphs should be presented in a standard way.

- The reviewer is unable to follow the paper goal.

- The figures need captures.

Reviewer #2: The scientific paper propose a detailed FE-based numerical analysis of one-piece zirconia implant concerning three clinically simulated models.

The authors present a very interesting study. This manuscript is well structured using a rich introduction concerning state of the art with respect to scientific literature overview.

The abstract is correct synthesised in order to understand the study goals and the principal obtained results. I would like to recommend this paper for publication in the journal “PLOS ONE“. The major drawbacks are itemized below:

Line 81: It is suggested to replace "The Authors" by "Marcián et al.".

Line 149: "direcrtion" ?

The type of the strain (true strain, equivalent strain, elastic strain, etc. ?) used in the entire manuscript should be specified.

Quality of the figures 3a-3b in pdf version of a manuscript is very poor. The values in legends are illegible.

Conclusions are not adequately supported by the data presented. It is suggested to add more conclusions based on the results found.

Reviewer #3: This is an interesting study, in which force transmission to bone was investigated in loaded Zir and Ti implants using finite element analysis models. Authors showed that loaded Zir implant transmit less force to bone compared to similar Ti implant, which is presumably to the differences in the stiffness between these two materials.

The experimental procedure is described in sufficient details. However, the statistical analysis is missing in the paper. It is not clear, how many experiments were made and how the differences in the transmission force were assessed.

Data on the force transmission should be presented as mean+/-SD (S.E.M).

In the results section, there are several sentences like “Strain on the implant surface for Ti-6Al-4V averaged at 4.00x10-4 and 1.78x10-4, respectively. These values were higher 69.39% and 62.32 respectively, than Zir which averaged at 1.94x10-4 and 9.33x10-5…”. Such constructions are little confusing, because it is not clear what is taken as 100 %. In this case, values for Ti are about two times higher than those for Zir, and therefore the difference could be estimated as about 100 %. This text, as well as similar passages in other places, should be rephrased. It is not necessary to mention the differences in %, because it can be estimated by readers themselves.

Minor comments

The quality of graphs is low, text is difficult to read. The resolution of figures should be improved.

I would recommend writing Ti6Al4V with the numbers in lowercase; the abbreviation Ti-6Al-4V is confusing, because the numbers seem to be related to wrong atoms.

6. PLOS authors have the option to publish the peer review history of their article (what does this mean?). If published, this will include your full peer review and any attached files.

Reviewer #1: No

Reviewer #2: No

Reviewer #3: No

---

## [Author Response · Author response to Decision Letter 0]

5 Jan 2020

Monolithic Yttria-Zirconia (Zir) dental implants promise superior esthetics while maintaining good biocompatibility. The purpose of this study was to evaluate the von Mises stress (MPa) and equivalent strain occurring around Zir implant using three clinically simulated finite element analysis (FEA) models for a missing maxillary central incisor. Two unidentified patients’ cone-beam computed tomography (CBCT) datasets with and without right maxillary central incisor were used to create the FEA models. Three different FEA models were made with bone structures that represent a healed socket (HS), reduced bone width edentulous site (RB), and immediate extraction socket with graft (EG). A one-piece abutment-implant fixture mimicking Straumann Standard Plus tissue level RN 4.1 X 11.8mm, for titanium alloy (Ti) and Zir were modeled. 178 N oblique load and 200 N vertical load were used to simulate occlusal loading. Von Mises stress and equivalent strain values for each implant model were measured within cortical and trabecular bone on the labial and palatal aspects. The FEA showed that the stress and strain distributions between the two materials had a statistically significant difference. Within the HS and RB models the labial-cervical region in the cortical bone exhibited highest stress, with Zir having statistically significant lower stress-strain means than Ti in both labial and palatal aspects. For the EG model the labial-cervical area had no statistically significant difference between Ti and Zir; however, Zir performed better than Ti against the graft. FEA models suggest that Ti, a more elastic material than Zir, contributes to the transduction of more overall forces to the socket compared to Zir. Thus, compared to Ti implants, Zir implants may be less prone to peri-implant bone overloading and subsequent bone loss in high stress areas especially in the labial-cervical region of the cortical bone. Zir implants may be more favorable in non-grafted normal edentulous or immediate extraction with grafting. This FEA modeling was completed on the basis to aid in conducting and validating future clinical studies.

---

## [Decision Letter · Decision Letter 1]

22 Jan 2020

PONE-D-19-30284R1

Finite Element Analysis of a One-piece Zirconia Implant in Anterior Single Tooth Implant Applications

PLOS ONE

Dear Dr. Bencharit,

Thank you for submitting your manuscript to PLOS ONE. After careful consideration, we feel that it has merit but does not fully meet PLOS ONE’s publication criteria as it currently stands. Therefore, we invite you to submit a revised version of the manuscript that addresses the points raised during the review process.

Please, take into consideration the comments made by Reviewer 1, and address the recommendations made by the reviewer. 

We would appreciate receiving your revised manuscript by Mar 07 2020 11:59PM. To enhance the reproducibility of your results, we recommend that if applicable you deposit your laboratory protocols in protocols.io, where a protocol can be assigned its own identifier (DOI) such that it can be cited independently in the future. For instructions see: http://journals.plos.org/plosone/s/submission-guidelines#loc-laboratory-protocols

We look forward to receiving your revised manuscript.

Kind regards,

Antonio Riveiro Rodríguez, PhD

Academic Editor

PLOS ONE

Reviewers' comments:

Reviewer's Responses to Questions

**Comments to the Author**

1. If the authors have adequately addressed your comments raised in a previous round of review and you feel that this manuscript is now acceptable for publication, you may indicate that here to bypass the “Comments to the Author” section, enter your conflict of interest statement in the “Confidential to Editor” section, and submit your "Accept" recommendation.

Reviewer #1: (No Response)

Reviewer #2: All comments have been addressed

Reviewer #3: All comments have been addressed

2. Is the manuscript technically sound, and do the data support the conclusions?

Reviewer #1: Partly

Reviewer #2: Yes

Reviewer #3: Yes

3. Has the statistical analysis been performed appropriately and rigorously? 

Reviewer #1: Yes

Reviewer #2: N/A

Reviewer #3: Yes

4. Have the authors made all data underlying the findings in their manuscript fully available?

Reviewer #1: Yes

Reviewer #2: Yes

Reviewer #3: Yes

5. Is the manuscript presented in an intelligible fashion and written in standard English?

Reviewer #1: Yes

Reviewer #2: Yes

Reviewer #3: Yes

6. Review Comments to the Author

Reviewer #1: The paper still needs a major revision. It should be mentioned that the author should addressed and replied to the reviewer comments carefully, because the comments are just to improve the paper. In the following, there are comments that are appropriate to reinforce the article.

**Abstract**

- Abstract is too long.

- Some of the most important research results should be mentioned at the end of this section.

- The authors mentioned “This FEA modeling was completed on the basis to aid in conducting and validating future clinical studies”. The reviewer think this is not a result.

**Keywords**

- Each paper needs keywords. The keywords are the main technical words that used in paper. For example, you can use following keywords: Finite element analysis; One-piece Zirconia Implant; titanium alloy; Zir implant.

**Conclusion**

- This section should be rewritten. It is suggested to use at least 11 line to explain the conclusion. Reader usually check the abstract and conclusion.

**General comments**

- The subtitles should be numbered.

- The quality of the figures should be increased.

- The legend of figure 3 is not clear. Most of the paper use figures with wide of 3.5 inch, or 7 inch. It is recommended to check the quality of figures in this size in your paper.

- The graphs should be presented in a standard way, meaning that the format of the legend, horizontal and vertical axis should be clear with high quality. If you are working with EXEL, please go to the dimension part and use 3.5inx3.5in for your graph. A standard file is attached for instance.

- The figures need captures.

- In vertical axis of the graphs, unite for stress is MPa, and it is not MPA.

Reviewer #2: (No Response)

Reviewer #3: (No Response)

7. PLOS authors have the option to publish the peer review history of their article (what does this mean?). If published, this will include your full peer review and any attached files.

Reviewer #1: No

Reviewer #2: No

Reviewer #3: Yes: Oleh Andrukhov

---

## [Author Response · Author response to Decision Letter 1]

27 Jan 2020

RESPONSES TO REVIEWERS

Reviewer #1: The paper still needs a major revision. It should be mentioned that the author should addressed and replied to the reviewer comments carefully, because the comments are just to improve the paper. In the following, there are comments that are appropriate to reinforce the article.

RESPONSE: We appreciate the time and thorough review. 

**Abstract**

- Abstract is too long.

RESPONSE: Abstract was shortened. 

- Some of the most important research results should be mentioned at the end of this section.

RESPONSE: We agreed. The end of the abstract is now read.

“Zir implants respond to occlusal loading differently then Ti implants. Zir implants may be more favorable in non-grafted edentulous or immediate extraction with grafting.”

- The authors mentioned “This FEA modeling was completed on the basis to aid in conducting and validating future clinical studies”. The reviewer think this is not a result.

RESPONSE: The sentence was removed.

**Keywords**

- Each paper needs keywords. The keywords are the main technical words that used in paper. For example, you can use following keywords: Finite element analysis; One-piece Zirconia Implant; titanium alloy; Zir implant.

RESPONSE: We agreed. The following keywords were added.

“Finite Element Analysis; One-piece Zirconia Implant; Titanium Alloy; Zirconia Implant.”

**Conclusion**

- This section should be rewritten. It is suggested to use at least 11 line to explain the conclusion. Reader usually check the abstract and conclusion.

RESPONSE: Additional information was added in the conclusion to complement the Abstract. The conclusion is now read:

“In general Zir implants behaves more favorable then Ti implants in terms of peri-implant stress distributions. Three different FEA models, healed edentulous site (HS), vertical periodontal defect under compression (RB), and immediate extraction with bone grafting site (EG), mimicking the common clinical scenarios suggested the following conclusion. Due to the stiffness of the material and its intrinsically high elastic modulus, Zir implants transmit less von Mises stress and induce lower equivalent strain to the peri-implant bone compared to Ti implants. This was statistically significant when cervical cortical bone (HS and RB models) and graft (EG model) were evaluated for one-piece Zir implant compared to its Ti counterpart. Therefore, the peri-implant bone surrounding Zir implants may be less prone to mechanically induced biologic peri-implant bone resorption. Zir implants may be considered for not only for its esthetic properties, but also the stress modulation properties of the material.”

**General comments**

- The subtitles should be numbered.

RESPONSE: There was no subtitle in this manuscript.

- The quality of the figures should be increased.

RESPONSE: We improved the figure resolution as suggested.

- The legend of figure 3 is not clear. Most of the paper use figures with wide of 3.5 inch, or 7 inch. It is recommended to check the quality of figures in this size in your paper.

RESPONSE: We improved the figure resolution and size as suggested.

- The graphs should be presented in a standard way, meaning that the format of the legend, horizontal and vertical axis should be clear with high quality. If you are working with EXEL, please go to the dimension part and use 3.5inx3.5in for your graph. A standard file is attached for instance.

RESPONSE: We improved the graph format resolution and added the legends as suggested.

- The figures need captures.

RESPONSE: Captures were added into the Figures per request.

- In vertical axis of the graphs, unite for stress is MPa, and it is not MPA.

RESPONSE: Changes have been made.

---

## [Editor Report · Decision Letter 2]

5 Feb 2020

Finite Element Analysis of a One-piece Zirconia Implant in Anterior Single Tooth Implant Applications

PONE-D-19-30284R2

Dear Dr. Bencharit,

We are pleased to inform you that your manuscript has been judged scientifically suitable for publication and will be formally accepted for publication once it complies with all outstanding technical requirements.

With kind regards,

Antonio Riveiro Rodríguez, PhD

Academic Editor

PLOS ONE

Additional Editor Comments (optional):

Please, revise the manuscript prior to its publication. English language should be revised. Some corrections are needed, e.g.: 

(Line 21, Abstract) Please, replace "...around Monolithic Yttria-Zirconia..." with "...around monolithic yttria-zirconia..."

(Line 38, Abstract) Please, replace "...differently then Ti implants." with "...differently than Ti implants."

(Line 299, Conclusions) Please, replace "In general Zir implants behaves more..." with "In general, Zir implants behave more..."

(Lines 308-309) Please, replace "Zir implants may be considered for not only for its esthetic properties, but also the stress modulation properties of the material." with "Zir implants may be considered not only due to its esthetic properties, but also due to the stress modulation properties of the material."

---

## [Editor Report · Acceptance letter]

7 Feb 2020

PONE-D-19-30284R2 

Finite Element Analysis of a One-piece Zirconia Implant in Anterior Single Tooth Implant Applications 

Dear Dr. Bencharit:

I am pleased to inform you that your manuscript has been deemed suitable for publication in PLOS ONE. Congratulations! Your manuscript is now with our production department. 

With kind regards,

on behalf of

Dr. Antonio Riveiro Rodríguez 

Academic Editor

PLOS ONE